# IS MY ACTION POLICY SAFE? POLIC3 TO THE RESCUE

## ABSTRACT

The use of machine learning in sequential decision-making tasks has grown substantially, intensifying concerns regarding the safety of learned policies and motivating research on policy verification. We present a new policy verification method based on the well-known IC3 algorithm. Unlike existing approaches, ours decouples reasoning about policy decisions from reasoning about the effects of these decisions on the environment in which the policy is executed. This separation allows us to leverage the latest advances in machine learning certification tools to handle the former subproblem, whilst relying on specialized solvers for the latter. Experiments confirm that our approach scales better and supports a wider variety of policy architectures than current state-of-the-art methods.

## 1 INTRODUCTION

Sequential decision-making is a central problem in artificial intelligence, concerned with choosing a course of actions that accomplishes a given objective. In the past decade, data-driven learning approaches such as reinforcement learning (RL) have achieved remarkable success in this field, surpassing human performance in a multitude of complex tasks in areas like game playing (e.g., Silver et al., 2016), finance (e.g., Yang et al., 2020), and robotics (e.g., Singh et al., 2022).

Despite this success, achieving reliable, predictable, and safe behavior remains a major challenge and a barrier to real-world deployment (e.g., García & Fernández, 2015; Chan et al., 2020; Giannaros et al., 2023). The learned action policies–often represented as deep neural networks–are complex, non-linear functions that operate as black boxes. Although they may exhibit strong performance according to training statistics, their behaviour may not align perfectly with the desired objective. Moreover, their response to novel or unexpected situations is *a priori* unclear, and there is generally no guarantee that they will adhere to critical safety constraints.

A growing line of research tackling these issues focuses on *verifying* that a learned policy meets desired behavior specifications under all circumstances (Bacci & Parker, 2022; Tambon et al., 2022; Vinzent et al., 2022; Abate et al., 2022; Schilling et al., 2023; Gross et al., 2023; Jain et al., 2024; Rober et al., 2024). To reason about the consequences of successive action decisions in a rigorous, exhaustive, and efficient manner, these verification methods leverage traditional model-checking techniques operating on a symbolic model of the environment that succinctly represents the (exponentially larger) state space. To integrate a learned policy into such techniques, existing works use specialized solvers capable of simultaneously reasoning over individual policy decisions and over their effects on the environment in a tightly coupled manner (Vinzent et al., 2022; Jain et al., 2024). This restricts their applicability to policy function architectures (e.g. feed forward neural networks with certain activation functions) that the specialized solvers support. Moreover, despite significant progress (e.g., Wu et al., 2024), the scalability of these solvers remains a major issue.

In this paper, we consider the verification of safety properties of the form "can executing my policy from an environment state satisfying formula $\phi_S$ ever reach an environment state satisfying formula $\phi_R$?". Building on the well-known hardware verification algorithm IC3 (Bradley, 2011), we introduce a new policy verification method which *decouples* the reasoning about the policy from the reasoning about the environment model, resulting in greater scalability and generality. This is made possible thanks to the following key contributions:

- We carefully identify and separate four core methods of the IC3 algorithm, yielding a generic verification algorithm that can be instantiated to diverse contexts. We analyze properties of the four methods that preserve the correctness of the overall algorithm.

- We instantiate this generic IC3 version to policy verification. A key step towards this goal is efficiently handling the so called *frame transition problem*, which asks whether, given two constraints $r$ and $\mathcal{F}$ identifying environment source and target states, the policy can cause a transition from $r$ to $\mathcal{F}$ under the environment model. We develop sound approximations of this problem and show how to efficiently solve these approximations with the help of state-of-the-art neural network and tree-ensemble certification tools (e.g., Zhang et al., 2018; Tjeng et al., 2019; Devos et al., 2021; Xu et al., 2021).

We empirically evaluate our new method on both existing and new benchmarks over a collection of feed-forward neural network and decision-tree ensemble policies. The results demonstrate that it is more efficient and often solves more problems than the current state of the art. Moreover, experimenting with policies produced by recent generalized planning approaches, namely ASNets (Toyer et al., 2020), we show that our method is capable of handling policies with complex function architectures that were previously out of reach.

## 2 BACKGROUND

### 2.1 MARKOV DECISION PROCESSES

We consider sequential decision-making problems in the form of Markov decision processes (MDPs). An MDP is a tuple $\mathcal{M} = \langle \mathcal{S}, \mathcal{A}, \mathcal{P}, \mu_0, \mathcal{R}, \gamma \rangle$, where $\mathcal{S}$ is a set of **states**, $\mathcal{A}$ is a set of **actions**, $\mathcal{P} : S \times \mathcal{A} \mapsto Dist(\mathcal{S})$ is the **transition-probability function**, $\mu_0 \in Dist(\mathcal{S})$ is the **initial state distribution**, $\mathcal{R} : \mathcal{S} \times \mathcal{A} \mapsto \mathbb{R}$ is the **reward function**, and $\gamma \in (0, 1]$ is the **discount factor**. For a state $s \in \mathcal{S}$, $\mathcal{A}(s) \subseteq \mathcal{A}$ denotes the actions **applicable** in $s$, where $a \in \mathcal{A}(s)$ if $\mathcal{P}(s'|s, a) > 0$ for some state $s'$. We assume for simplicity that $\mathcal{A}(s) \neq \emptyset$. A **policy** is a function $\pi : \mathcal{S} \mapsto \mathcal{A}$ mapping each state $s \in \mathcal{S}$ to an applicable action $\pi(s) \in \mathcal{A}(s)$. The **value** of $\pi$ in the state $s$ is the expected discounted cumulative reward when executing $\pi$ from $s$. An optimal solution of $\mathcal{M}$ is a policy maximizing the expected value for the initial states. Even if $\mathcal{M}$ is not known explicitly, approximately optimal policies can be computed automatically via reinforcement learning (RL) (Arulkumaran et al., 2017) or imitation learning (IL) (Hussein et al., 2017). In the following, we consider policies represented by a parameterized function $f_\theta : \mathcal{S} \mapsto \mathbb{R}^{|\mathcal{A}|}$ (e.g., neural networks). $f_\theta$ induces the policy $\pi_\theta(s) := argmax_{a \in \mathcal{A}(s)} f_\theta(s)[a]$ that maps each state $s$ to the action ranked highest by $f_\theta$, while masking inapplicable actions.

### 2.2 POLICY VERIFICATION

The learning algorithms typically provide only statistical estimates of some performance metric for the delivered policy $\pi_\theta$. In contrast, policy verification rigorously analyzes $\pi_\theta$ based on a declarative specification of the environment, in order to obtain strict guarantees that the behaviour of $\pi_\theta$ fulfills desired properties that may or may not align with the reward objective.

We consider environment models in a guarded-command language. An environment model is a tuple $\mathcal{E} = \langle Vars, \mathcal{L}, \mathcal{C} \rangle$, where *Vars* is a set of **integer variables** $v \in$ *Vars* with bounded domains identified by lower and upper bounds $L_v, U_v \in \mathbb{Z}$, $\mathcal{L}$ is a set of **labels**, and $\mathcal{C}$ is a set of **(guarded) commands**. We call any Boolean combination of linear constraints over *Vars* with coefficients in $\mathbb{Z}$ a **linear condition**. A **guarded command** $c \in \mathcal{C}$ has the form $l_c : grd_c \triangleright eff_c$, where $l_c \in \mathcal{L}$ is the **label** of $c$, the **guard** $grd_c$ is a linear condition, and the **effect** $eff_c$ maps each variable to a linear expression over *Vars* with coefficients in $\mathbb{Z}$. A state of $\mathcal{E}$ assigns each variable in *Vars* to a value in that variable's domain. $\mathcal{S}_\mathcal{E}$ denotes the set of all states. A command $c$ is **applicable** in the state $s$ if $s \models grd_c$. Applying $c$ in $s$ results in the state $s[\![c]\!]$ defined such that for all $v \in$ *Vars*, $s[\![c]\!](v) := \chi_v[s]$, where $\chi_v = eff_c(v)$ is the linear expression assigned by $c$ to $v$, and $\chi_v[s]$ is the result of the evaluation of $\chi_v$ in $s$. A label $l$ is applicable in the state $s$, if a command $c$ with label $l_c = l$ is applicable in $s$. The applicable labels are denoted by $\mathcal{L}(s) \subseteq \mathcal{L}$. The semantics of the environment model $\mathcal{E}$ is captured via the **transition system** $\Theta_\mathcal{E} = \langle \mathcal{S}_\mathcal{E}, \mathcal{L}, \mathcal{T}_\mathcal{E} \rangle$ with transitions $\mathcal{T}_\mathcal{E} := \{\langle s, l_c, s[\![c]\!]\rangle \mid s \in \mathcal{S}_\mathcal{E}, c \in \mathcal{C}, s \models grd_c\}$. A **path** in $\Theta_\mathcal{E}$ is a sequence $s_0, l_0, s_1, l_1, \ldots, s_n$ so that $\langle s_i, l_i, s_{i+1}\rangle \in \mathcal{T}_\mathcal{E}$ holds for all $i \in \{0, \ldots, n-1\}$. A **safety property**, or simply property, is a pair $P = \langle \phi_S, \phi_R \rangle$ consisting of a linear **start condition** $\phi_S$ and a linear **reach condition** $\phi_R$. $P$ is satisfied in a transition system $\Theta$, if $\Theta$ contains a path from a state satisfying $\phi_S$ to a state satisfying $\phi_R$. An environment model $\mathcal{E}$ satisfies $P$, if $P$ is satisfied in the transition system

$\Theta_{\mathcal{E}}$, i.e., if there is a path $s_0, a_0, s_1, a_1, \ldots, s_n$ such that (i) $s_0 \models \phi_S$, and (ii) $s_n \models \phi_R$, and (iii) $\forall i \in \{0, \ldots, n-1\} : \langle s_i, a_i, s_{i+1} \rangle \in \mathcal{T}_{\mathcal{E}}$.

A policy for $\mathcal{E}$ is a function $\pi : \mathcal{S}_{\mathcal{E}} \mapsto \mathcal{L}$ such that $\pi(s) \in \mathcal{L}(s)$ for all states $s \in \mathcal{S}_{\mathcal{E}}$. Let $\pi_\theta : \mathcal{S} \mapsto \mathcal{A}$ be a policy trained based on the MDP $\mathcal{M}$. We assume for simplicity that $\pi_\theta$ is a policy for the model $\mathcal{E}$. In particular, $\mathcal{S} = \mathcal{S}_{\mathcal{E}}$ and $\mathcal{A} = \mathcal{L}$, i.e., the policy's input and output match the states and labels of the environment model. We remark, however, that this does not impose any restriction as long as it is possible to represent the necessary interface functions, translating between $\mathcal{M}$ and $\mathcal{E}$, in the function space of $\pi_\theta$. The behavior of $\pi_\theta$ in $\mathcal{E}$ is formally defined as the transition sub-system $\Theta_{\mathcal{E}}^{\pi_\theta} = \langle \mathcal{S}_{\mathcal{E}}, \mathcal{A}, \mathcal{T}_{\mathcal{E}}^{\pi_\theta} \rangle$ with the transitions $\mathcal{T}_{\mathcal{E}}^{\pi_\theta} \subseteq \mathcal{T}_{\mathcal{E}}$, where $\langle s, a, s' \rangle \in \mathcal{T}_{\mathcal{E}}^{\pi_\theta}$ if $\pi_\theta(s) = a$. $\pi_\theta$ **satisfies** the property $P$ if $P$ is satisfied in the transition sub-system $\Theta_{\mathcal{E}}^{\pi_\theta}$, i.e., if there is path $s_0, a_0, s_1, a_1, \ldots, s_n$ such that (i) $s_0 \models \phi_S$, and (ii) $s_n \models \phi_R$, and (iii') $\forall i \in \{0, \ldots, n-1\} : \langle s_i, a_i, s_{i+1} \rangle \in \mathcal{T}_{\mathcal{E}}^{\pi_\theta}$. Note the subtle but important difference between the conditions (iii) and (iii'). In the latter, one is interested in the specific transition choices made by the given policy only. That difference apart, verifying whether $\pi_\theta$ satisfies a property $P$ inherits the worst-case **PSPACE**-complete complexity from the model verification problem (Demri & Schnoebelen, 1998).

### 2.3 Neural-Network and Tree-Ensemble Certification

Certifying the decisions of learned neural networks and tree ensembles has become a standard problem in the machine learning literature. In the following, we will leverage such certification methods in order to verify efficiently properties of learned policies. We consider two approaches specifically. To handle neural network policies, we consider **LiRPA** (Zhang et al., 2018; Singh et al., 2019). Let $f_\theta : \mathbb{R}^n \mapsto \mathbb{R}$ be a neural network mapping an $n$-dimensional input to a single real number. Moreover, let $\Phi$ denote an interval constraint $L \leq x \leq U$ where $L, U \in \mathbb{R}^n$; $x$ denoting the input to $f_\theta$. LiRPA is a state-of-the-art method certifying that

$$\forall x \in \mathbb{R}^n, x \models \Phi : f_\theta(x) \geq 0. \tag{1}$$

For tree ensembles, we rely on **Veritas** (Devos et al., 2021). Let $g_\theta : \mathbb{R}^n \mapsto \mathbb{R}^m$ be a tree ensemble mapping an $n$-dimensional input to scores of $m$ different classes, and $\Phi$ be an interval constraint as before. Let $C$ denote the index of a target class. Then, amongst other variations, Veritas is an efficient tool for certifying whether

$$\forall x \in \mathbb{R}^n, x \models \Phi : g_\theta(x)[C] < \max_{C' \neq C} g_\theta(x)[C'], \tag{2}$$

i.e., that no input identified by $\Phi$ is classified as $C$.

## 3 Method

We extend the reach of the IC3 algorithm to policy verification. In Sec. 3.1, we introduce a generic version of the IC3 algorithm, adapted from the literature (Bradley, 2011; Eén et al., 2011). This generic version leaves open the implementation of core functions, and depending on their implementation, the algorithm can be adapted to different settings. In Sec. 3.2, we will instantiate the functions for verifying environment models. This lays down the basis for our development of IC3 for policy verification in Sec. 3.3. For space reasons, we describe only central parts for understanding the algorithm. Further details are provided in Appendix A.

### 3.1 Generic IC3 Algorithm

Let $\Theta = \langle \mathcal{S}, \mathcal{A}, \mathcal{T} \rangle$ be a transition system, and $P = \langle \phi_S, \phi_R \rangle$ be a property. We next present our IC3 variant for verifying whether $\Theta$ satisfies $P$. $\Theta$ is accessed only implicitly via the sub-procedures, so is treated by the general algorithm like a black box and does not need to be provided directly.

Alg. 1 shows the pseudocode. We assume for simplicity that no state satisfies both $\phi_S$ and $\phi_R$, in which case $P$ is trivially satisfiable. IC3 incrementally builds path-length dependent reachability information in the form of linear conditions, called **frames**, $\mathcal{F}_0, \mathcal{F}_1, \ldots, \mathcal{F}_N$. All frames but $\mathcal{F}_0$ are conjunctions of *clauses* of the form $\neg r$, where $r \subseteq s$ is a partial variable assignment derived from a state $s$. We describe how $r$ is computed below. With some abuse of notion, we treat frames as sets of such clauses. The frames are constructed in such a way that the $n$-th frame $\mathcal{F}_n$ identifies a necessary condition for a state to have a path of length $n$ or less to some state satisfying $\phi_R$. $\mathcal{F}_0$

**Input** : Implicitly defined transition system $\Theta = \langle \mathcal{S}, \mathcal{A}, \mathcal{T} \rangle$, property $P = \langle \phi_S, \phi_R \rangle$
**Output:** *true* if $P$ is satisfied in $\Theta$ and *false* otherwise

1   $\mathcal{F}_0 \leftarrow \phi_R$; $\mathcal{F}_1 \leftarrow \emptyset$; $N \leftarrow 1$ ;        `// initialize frames and path-length limit`
2   **while** *forever* **do**
3     $s_{\text{start}} \leftarrow \texttt{selectStartState()}$ ;     `// get state` $s \in \mathcal{S}$ `s.t.` $s_{\text{start}} \models \phi_S$ `and` $s_{\text{start}} \models \mathcal{F}_N$
4     **if** *selection not possible* **then**
5       $\mathcal{F}_{N+1} \leftarrow \emptyset$;                    `// open new frame`
6       $\texttt{propagateClauses()}$ ;     `// try pushing clauses from` $\mathcal{F}_m$ `into` $\mathcal{F}_{m+1}$`, for all` $m$
7       **if** $\mathcal{F}_N = \mathcal{F}_{N+1}$ **then**   **return false** ;            `//` $\phi_R$ `is not reachable`
8       **else** $N \leftarrow N + 1$; **continue** ;       `// increase path length limit and reselect` $s_{\text{start}}$
9     **end**
      `// find path of length` $N$ `from` $s_{\text{start}}$ `to some` $s_N \models \mathcal{F}_0$ `in` $\Theta$`, or remove` $s_{\text{start}}$ `from` $\mathcal{F}_N$
10     $Q \leftarrow$ empty queue; insert $\langle s_{\text{start}}, N \rangle$ into $Q$;
11     **while** $Q$ *is not empty* **do**
12       $\langle s, n \rangle \leftarrow$ pop element with minimal $n$ from $Q$;
13       **if** $n = 0$ **then**   **return true**;        `// found a path from` $s_{\text{start}} \models \phi_S$ `to` $s_N \models \phi_R$ ;
14       $s' \leftarrow \texttt{selectSuccessorState}(s, n - 1)$ ;     `// get transition` $\langle s, a, s' \rangle$ `s.t.` $s' \models \mathcal{F}_{n-1}$
15       **if** *selection not possible* **then**
        `// compute small reason` $r \subseteq s$ `for the absence of such a transition`
16         $r \leftarrow \texttt{generalizeReason}(s, n - 1)$;
17         **foreach** $m = 1, \ldots, n$ **do** $\mathcal{F}_m \leftarrow \mathcal{F}_m \cup \{\neg r\}$ ;
18       **else**
19         insert $\langle s, n \rangle$ into $Q$ ;       `// allow revisiting` $s$ `if one later backtracks from` $s'$
20         insert $\langle s', n - 1 \rangle$ into $Q$ ;           `// continue with` $s'$
21       **end**
22     **end**
23 **end**

**Algorithm 1:** Generic IC3 algorithm checking whether $P$ is satisfied in $\Theta$. $\Theta$ is accessed by sub-procedures only. The sub-procedures are deliberately left open, see text.

is set to $\phi_R$, while the other frames are generated during the execution of IC3. The purpose of the main loop (line 2) is finding a path $s_0 = s_{\text{start}}, a_0, s_1, a_1, \ldots, s_N$ in $\Theta$ from a start state $s_{\text{start}} \models \phi_S$ to a state satisfying the desired reach condition $s_N \models \phi_R$.

Given the mentioned property of the frames, every such path must necessarily traverse the frame segments in reverse direction, i.e., it must hold for all $n \in \{0, \ldots, N\}$ that $s_n \models \mathcal{F}_{N-n}$. The frames can hence be used to guide the search for the path. The $\texttt{selectStartState}$ procedure initiates path construction by selecting the start state with $s_{\text{start}} \models \mathcal{F}_N$ directly (line 3).

**Assertion 1.** $\texttt{selectStartState()}$ *returns* $s \in \mathcal{S}$ *s.t.* $s \models \phi_S \wedge \mathcal{F}_N$, *or throws an error.*

The inner loop (line 11) then incrementally extends the current path prefix $s_0, a_0, \ldots, s_n$, for $n = 0, \ldots, N - 1$, by finding a transition from $s_n$ to some $s_{n+1}$ that is one step closer to $\phi_R$:

**Assertion 2.** $\texttt{selectSuccessorState}(s, n - 1)$ *returns* $s' \in \mathcal{S}$ *such that* $s' \models \mathcal{F}_{n-1}$ *and there is a transition* $\langle s, a, s' \rangle \in \mathcal{T}$, *for some* $a \in \mathcal{A}$, *or throws an error.*

The construction process might however fail at two points. First, there might be no state $s_{\text{start}}$ satisfying $s_{\text{start}} \models \phi_S \wedge \mathcal{F}_N$ (line 4). If this is the case, then, as per the frame definition, the start states cannot reach $\phi_R$ within the length limit of $N$. Ignoring the additional termination check for the moment, IC3 *opens* a new frame $\mathcal{F}_{N+1}$, initialized so that it is satisfied by all states, and tries strengthening the frames by moving clauses from lower to higher frames:

**Assertion 3.** $\texttt{propagateClauses()}$ *moves a clause* $\neg r \in \mathcal{F}_m$ *into* $\mathcal{F}_{m+1}$ *only if no state satisfying* $r$ *has a transition into* $\mathcal{F}_m$, *i.e., is at least* $m + 1$ *steps away from* $\phi_R$. *Formally,* $s \models r$ *must imply that* $s' \not\models \mathcal{F}_m$, *for all* $\langle s, a, s' \rangle \in \mathcal{T}$.

The main loop restarts with the increased length limit $N + 1$.

Secondly, since the frame $\mathcal{F}_n$ may over-approximate the $n$-steps bounded reachability of $\phi_R$, it may happen that the incumbent state $s \models \mathcal{F}_n$ (from line 12) does actually not have any path to $\phi_R$ with length of at most $n$. In particular, the desired transition $\langle s, a, s' \rangle$ to $s' \models \mathcal{F}_{n-1}$ might not exist (line 15). This situation causes a frame refinement. To this end, $\texttt{generalizeReason}$ distills a small **reason** $r \subseteq s$ for when at least $n + 1$ steps are necessary to reach $\phi_R$.

**Assertion 4.** `generalizeReason(s, n − 1)` *returns a partial variable assignment $r \subseteq s$ that (i) entails $\neg\phi_R$, i.e., for all states $t$, if $t \models r$, then $t \not\models \phi_R$, and (ii) is only satisfied by states without transition into $\mathcal{F}_{n-1}$, i.e., if $t \models r$, then $t' \not\models \mathcal{F}_{n-1}$ holds for all $\langle t, a, t' \rangle \in \mathcal{T}$.*

(i) ensures that the states satisfying $r$ are at least one step away from $\phi_R$. With the absence of transitions into $\mathcal{F}_{n-1}$ as per (ii), the states satisfying $r$ can hence not reach $\phi_R$ in $n$ steps. This information is incorporated into the algorithm by adding $\neg r$ to the corresponding frames. In particular, $s \not\models \mathcal{F}_n$ is true after the refinement. Note that for the effectiveness of the overall algorithm, it is absolutely essential that the reasons generalize, i.e., ruling out many states at once in a single refinement step.

After the refinement, the search resumes at the previous state looking for a successor $s'$ that satisfies the refined frame $\mathcal{F}_n$. The backtracking may then further continue. The main loop repeats until the desired path is found, showing the satisfaction of $P$. In order to not repeat this process indefinitely if $P$ is unsatisfiable, a convergence check is conducted in line 7. Once $\mathcal{F}_N = \mathcal{F}_{N+1}$, IC3 has computed a condition that is invariant under transitions and separates the start states $\phi_S$ from $\phi_R$. In other words, IC3 has found an unsatisfiability proof. To ensure that $\mathcal{F}_N = \mathcal{F}_{N+1}$ is guaranteed to hold eventually, we must make sure that redundant knowledge can be added to the frames without affecting the syntactic equality check:

**Assertion 5.** *If* `generalizeReason(s, n − 1)` *returns $r$ and $\neg r \in \mathcal{F}_m$ for some $m \leq n - 1$,* `propagateClauses` *would have moved $\neg r$ into $\mathcal{F}_n$.*

**Theorem 1.** *Provided that the sub-procedures guarantee the stated properties, Alg. 1 terminates and returns* true *if and only if the property $P$ is satisfied in the transition system $\Theta$.*

### 3.2 IC3 FOR ENVIRONMENT MODEL VERIFICATION

Let $\mathcal{E} = \langle \textit{Vars}, \mathcal{A}, \mathcal{C} \rangle$ be an environment model. To verify whether $\mathcal{E}$ satisfies the property $P$ using IC3, we need to provide implementations of the four sub-procedures of Alg. 1 for the transition system $\Theta_\mathcal{E}$. Importantly, in order for the algorithm to be efficient, it is crucial that the implementations operate on the description of $\mathcal{E}$ itself rather than on $\Theta_\mathcal{E}$ directly. In the original setting (Bradley, 2011; Eén et al., 2011), where the model was described in propositional logic, the backbone to this end were SAT solvers (Biere et al., 2021). Here, we instead consider constraint systems over integer variables with linear constraints, which can be solved via SMT (Barrett et al., 2021).

`selectStartState` can be straightforwardly implemented by representing the requirements of Assert. 1 as a constraint system, and getting a solution to that system from an SMT solver.

The three remaining sub-procedures reason over the model's transitions $\mathcal{T}_\mathcal{E}$. Their implementation commonly requires a method for deciding decision problems of the following general form

**Definition 1.** *Let $r$ be a (partial) variable assignment, and let $n \in \{0, \dots, N\}$. The* frame transition problem *for $r$ and $n$ is*

$$\exists s \in \mathcal{S}_\mathcal{E} : \exists a \in \mathcal{A} : \exists s' \in \mathcal{S}_\mathcal{E} : \quad s \models r \ \wedge \ s' \models \mathcal{F}_n \ \wedge \ \langle s, a, s' \rangle \in \mathcal{T}_\mathcal{E} \qquad (3)$$

In words, this decision problem asks whether it is possible to transition into the frame $\mathcal{F}_n$ if $r$ is satisfied. Importantly, it can be formulated as the SMT problem $\mathsf{FrameTransition}[r, n]$, without having to enumerate $\mathcal{S}_\mathcal{E}$ and $\mathcal{T}_\mathcal{E}$ explicitly. The encoding follows that in (Eén et al., 2011).

`selectSuccessorState` and `propagateClauses` can be implemented using $\mathsf{FrameTransition}$ in a straightforward manner. To implement `generalizeReason(s, n − 1)`, we follow a greedy state minimization procedure as in earlier works (Bradley, 2011; Eén et al., 2011). We initialize the reason to $r := s$, which is guaranteed to satisfy (i) and (ii) of Assert. 4. Afterwards, we iterate over all variables $v \in \textit{Vars}$, checking whether $r \setminus \{v \mapsto r(v)\}$ still satisfies (i) and (ii) using $\mathsf{FrameTransition}$. If yes, we update $r$ accordingly. If no, we skip directly to the next variable. The resulting $r$ obviously guarantees Assert. 4.

### 3.3 IC3 FOR POLICY VERIFICATION

Let $\pi_\theta : \mathcal{S}_\mathcal{E} \mapsto \mathcal{A}$ be a neural-network or tree-ensemble policy. To analyze $\pi_\theta$, we have to instantiate the four sub-procedures of Alg. 1 for the transition sub-system $\Theta_\mathcal{E}^{\pi_\theta}$. Since `selectStartState` is completely independent of the policy, its implementation from Sec. 3.2 can be used as is. To implement the remaining sub-procedures, we first adapt the decision problem from Def. 1 to policies, and develop a method for solving it efficiently.

### 3.3.1 POLICY FRAME TRANSITION PROBLEM

Recall the decision problem from Def. 1. To handle policies, the problem changes slightly:

**Definition 2.** *Let $r$ and $n$ be as before. The* policy frame transition problem *for $r$ and $n$ is*

$$\exists s \in \mathcal{S}_\mathcal{E} : \exists a \in \mathcal{A} : \exists s' \in \mathcal{S}_\mathcal{E} : \quad s \models r \ \wedge \ s' \models \mathcal{F}_n \ \wedge \ \langle s, a, s' \rangle \in \mathcal{T}_\mathcal{E} \ \wedge \ \pi_\theta(s) = a \quad (4)$$

Compared to (3), (4) includes the requirement that the policy chooses the action of the desired transition. This subtle difference unfortunately complicates solving this decision problem tremendously.

In principle, it is possible to extend the previous SMT-based approach to solve (4), provided that the policy function $\pi_\theta$ allows compiling the condition $\pi_\theta(s) = a$ into SMT constraints. For feed-forward neural networks with relu activation units and decision trees, this is possible (e.g., Tjeng et al., 2019; Ceccon et al., 2022). Such a direct encoding however has two major disadvantages. First, the encoding is possible only for limited families of functions. Secondly, the encoding has to mimic the function structure of $\pi_\theta$, which can significantly increase the size of the SMT problem even up to the point where solving it becomes completely unpractical already for small neural networks and decision trees (cf. e.g., Xu et al., 2021; Vinzent et al., 2022; Jain et al., 2024). While in recent years, some research was spent on SMT solvers that include dedicated neural network reasoning methods, scalability remains a major bottleneck (e.g., Wu et al., 2024).

### 3.3.2 APPROXIMATING THE POLICY FRAME TRANSITION PROBLEM

To avoid the mentioned deficiencies, we abstain from solving the decision problem exactly, instead decomposing (4) into separate transition and action selection parts. More specifically, after moving the action quantification to the front, we split the remaining inner condition as follows:

$$\exists a \in \mathcal{A} : \quad (5.1) \ \exists s, s' \in \mathcal{S}_\mathcal{E} : \ s \models r \ \wedge \ s' \models \mathcal{F}_n \ \wedge \ \langle s, a, s' \rangle \in \mathcal{T}_\mathcal{E} \text{ and} \qquad (5)$$
$$(5.2) \ \exists s \in \mathcal{S}_\mathcal{E} : \ s \models r \ \wedge \ \pi_\theta(s) = a$$

Note that due to the independent state quantifications in (5.1) and (5.2), this formulation is not equivalent to (4). However, it is easy to show that it constitutes a necessary condition:

**Theorem 2.** *If there is no action satisfying* (5)*, then also* (4) *is not satisfiable.*

The motivation behind separating conditions (5.1) and (5.2) is to enable using separate solvers dedicated for the different parts; in particular, without compiling the policy function $\pi_\theta$ into SMT. Let $a \in \mathcal{A}$. (5.1) can be checked via the SMT FrameTransition$[r, n, a]$, similar to Sec. 3.2.

In order to check condition (5.2) efficiently, we want to leverage the certification tools from Sec. 2.3. Let $f_\theta : \mathcal{S}_\mathcal{E} \mapsto \mathbb{R}^{|\mathcal{A}|}$ be the neural network or tree ensemble underlying $\pi_\theta$. By the definition of $\pi_\theta$, $\pi_\theta(s) = a$ is true if $f_\theta(s)[a] \geq \max_{a' \in \mathcal{A}(s):a' \neq a} f_\theta(s)[a']$. Unfortunately, the restriction to the applicable actions still induces a dependency on the environment model. So, (5.2) cannot be tackled by the certification tools directly. We further relax the condition by moving the applicable actions computation out of the equation. Specifically, let $\underline{A} \subseteq \mathcal{A}$ be an under-approximation of the applicable actions, i.e., such that $\underline{A} \subseteq \mathcal{A}(s)$ holds for all the states $s \in \mathcal{S}_\mathcal{E}$ with $s \models r$.

**Theorem 3.** *Consider the condition*

$$\forall s \in \mathcal{S}_\mathcal{E}, s \models r : f_\theta(s)[a] < \max_{a' \in \underline{A}, a' \neq a} f_\theta(s)[a'] \qquad (6)$$

*If* (6) *is satisfied, then $\pi_\theta(s) \neq a$ holds for all states $s \in \mathcal{S}_\mathcal{E}$ where $s \models r$, i.e.,* (5.2) *is not satisfiable.*

In words, (6) requires for all states satisfying $r$ that the score of $a$ is worse than that of an action in the under-approximation $\underline{A}$. The computation of the under-approximation $\underline{A}$ can be delegated to an SMT solver. Given $\underline{A}$, (6) can be compiled into the certification problems from Sec. 2.3, as described below. In summary, this leads to the following algorithm, called APFT $(r, n)$, approximating the policy frame transition problem (4). APFT iterates over all actions $a \in \mathcal{A}$. It checks whether $a$ satisfies (5.1) through SMT. If so, APFT obtains an applicable actions under-approximation $\underline{A}$, and checks whether $a$ satisfies (6) via the detour to the certification problem. If (6) is satisfiable, then (5.2) and therewith (4) is not. APFT continues with the next action. Otherwise, APFT returns true.

**Corollary 1.** APFT $(r, n)$ *returns false only if* (4) *is not satisfiable.*

**Tree ensembles** To solve (6) for the tree ensemble $f_\theta$ using VERITAS (Devos et al., 2021), we need to get rid of the restriction in the maximization of (6). To this end, we use a simple masking

| | FFNN | | | | DTE | | | ASNET | |
|---|---|---|---|---|---|---|---|---|---|
| | beluga (3) | blocks (24) | npuzzle (6) | transport (5) | beluga (6) | blocks (18) | transport (6) | blocks (116) | npuzzle (97) |
| BRFS | 0 | **17** | 4 | 3 | 0 | **16** | 2 | 33 | 71 |
| PPA | 0 | 14 | **6** | **4** | 2 | 13 | 4 | – | – |
| POLIC3-nog | 0 | 5 | 5 | 3 | 0 | 0 | 2 | 63 | **95** |
| POLIC3 | 0 | 14 | **6** | 3 | **4** | 15 | **6** | **96** | 93 |

Table 1: Coverage table (number of solved instances). Per benchmark domain best values are highlighted in bold. Results for the different policy types FFNN (feed-forward neural networks), DTE (decision-tree ensembles), and ASNET are separated. Total instance count shown in braces.

approach, constructing a tree ensemble $g_\theta$ that guarantees that (6) is satisfied iff $g_\theta$ satisfies the certification problem (2). $g_\theta$ is constructed by adding to $f_\theta$ the penalty term $-P_{\max}$ for all actions different from $a$ and $\underline{A}$. The desired relation holds for $P_{\max}$ such that $P_{\max} > f_\theta(s)[a']$ for all $s$ and $a'$, which can be computed based on the leaf values in the decision trees.

**Neural networks** For neural networks, we compile the problem into a neural network $g_\theta$ that satisfies the certification problem (1) iff (6) is not satisfied. $g_\theta$ is constructed by appending two additional layers to $f_\theta$. The first auxiliary layer has $|\underline{A}|$ outputs, returning $z_{a'} := \text{relu}(f_\theta[a'] - f_\theta[a])$ for all $a' \in \underline{A}$. The final layer returns $\sum_{a' \in \underline{A}} z_{a'} - \epsilon$ for a small $\epsilon > 0$. If $g_\theta(s) < 0$ then $\sum_{a' \in \underline{A}} z_{a'} < \epsilon$. Since $z_{a'} \geq 0$, it follows for all $a' \in \underline{A}$ that $f_\theta(s)[a] + \epsilon > f_\theta(s)[a']$. So, if $\epsilon$ is sufficiently small, (6) must be violated. Hence, $g_\theta$ satisfies (1) iff (6) is violated.

### 3.3.3 IMPLEMENTATION OF IC3 SUB-PROCEDURES

We finally have all tools ready to implement the three remaining sub-procedures of IC3. Instead of relying on the approximation of the policy frame transition problem, we implement `selectSuccessorState`$(s, n - 1)$ by searching for the desired state $s'$ directly. Note that the relevant transitions can be enumerated efficiently through a single iteration over the commands $C$. This simple procedure obviously satisfies Assert. 2, and avoids the SMT related problems.

Since `generalizeReason` and `propagateClauses` require solving (4) for conditions $r$ that are satisfied by potentially many states, this simple enumeration approach is unfortunately not feasible. We adopt the general procedures from Sec. 3.2, substituting the exact frame transition test by APFT. The use of the approximation in APFT does not affect Assert. 3 and 4. The resulting `propagateClauses` function satisfies Assert. 3 since it moves a clause $\neg r \in \mathcal{F}_m$ only into $\mathcal{F}_{m+1}$ if APFT$(r, m)$ returns false. It follows from Cor. 1 that there is no policy transition from $r$ into $\mathcal{F}_m$, as required. Similarly, for `generalizeReason`, recall that the state minimization process removes a variable assignment only if the frame transition test is false. Since by Cor. 1, APFT returns false only if (4) is not satisfiable, condition (ii) of Assert. 4 is preserved. Condition (i) is not affected by the use of APFT. Finally, Assert. 5 is satisfied since both functions rely on APFT.

## 4 EXPERIMENTAL EVALUATION

Our implementation, called POLIC3, is in C++ and supports the analysis of neural-network and tree-ensemble policies. Environment models are provided in the JANI guarded-command language (Budde et al., 2017). We re-implemented LiRPA (Zhang et al., 2018) for neural networks certification. To certify tree ensembles, we interface with the open-source tool VERITAS (Devos et al., 2021). We use the Z3 SMT solver (de Moura & Bjørner, 2008). The code is publicly available.[1]

Our experiments aim at answering the following main questions:

(Q1) How does POLIC3 compete against state-of-the-art approaches?

(Q2) Can POLIC3 cope with more complex policy architectures, such as those underlying state-of-the-art generalized planning policies?

---

[1]Link omitted to preserve anonymity; but code is available as supplemental material

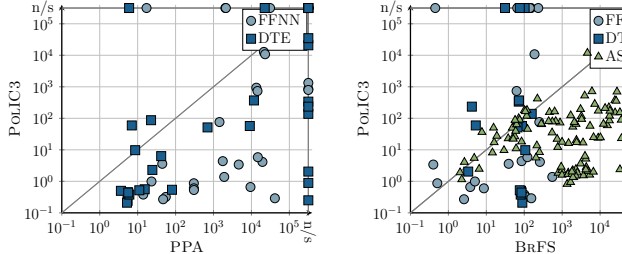

Figure 1: Per-instance runtime (in seconds) comparison of POLIC3 ($y$-axes) to PPA and BRFS ($x$-axes). "n/s" marks instances not solved in the resource limits.

**Benchmarks** We consider all benchmarks with integer-variable environments from prior work (Vinzent et al., 2022; Jain et al., 2024). This encompasses four different problem domains (beluga, blocks, npuzzle, transport) with a total of 72 benchmark instances, where 38 instances contain a feed-forward neural-network (FFNN) policy and 34 contain a decision-tree ensemble (DTE) policy. We further extended this collection by learning new policies in blocks and npuzzle using ASNET (Toyer et al., 2020), one of the state of the art approaches in generalized planning. ASNET constructs graph neural networks, leveraging a clever weight sharing scheme to instantiate policies for different planning problem instances. ASNET requires planning problem descriptions in PDDL (Fox & Long, 2003), not JANI. For training, we manually created the necessary PDDL encodings. The training instances are randomly generated. In order to verify the learned ASNET policies $\pi^{\text{ASNET}}$, we implemented translation functions $F$, mapping a JANI state into a PDDL state, and $G$, mapping a PDDL action into a JANI action, as feed-forward neural networks using relu activation units, and consider the concatenation $G \circ \pi^{\text{ASNET}} \circ F$ for verification. We created additional 116 benchmark instances in blocks and 97 instances in npuzzle in this manner.

**Baselines and configurations** We compare POLIC3 to policy predicate abstraction (PPA), the current state-of-the-art algorithm for policy verification (Vinzent et al., 2022; Jain et al., 2024). Moreover, as an additional baseline, we include an exhaustive search method (BRFS), which runs multiple breadth-first searches in $\Theta_{\mathcal{E}}$ to find a property satisfying path. It enumerates the set of start states and $\Theta_{\mathcal{E}}$ incrementally, and if the desired path is found, might terminate before either of them has been fully constructed. Finally, we also experiment with a POLIC3 variant, called POLIC3-nog, which does not use reason generalization, instead just using the entire state for the frame refinements.

**Setup** All experiments were run on Intel Xeon E5-2695 servers. Like in previous setups, each run was limited to a single CPU thread, 12 hours runtime, and 4 GB memory.

**Results** Tab. 1 compares the number of instances each method could solve. We do not have results for PPA on ASNET policies, because the implementation by Vinzent et al. (2022) does not support this type of policy, and adding that support is absolutely non-trivial. Comparing POLIC3 to the state-of-the-art policy verifier PPA, POLIC3 achieves the same or better coverage in all but one benchmark domains and across both remaining policy types. For FFNN policies, coverage is identical except for transport, where PPA is able to handle one more instance. For DTE policies, POLIC3 improves coverage in all three benchmark domains. In contrast to the DTE policies, for the FFNN policies, we observed that POLIC3's reason generalization method frequently fails to find small, and thus generalizing, reasons. We attribute this to a lack of consistent structure in the decisions made by the FFNN policies, owed to the way these policies were trained by Vinzent et al. (2022) (Q-learning). The importance of reason generalization becomes evident when disabling in POLIC3 the generalization method entirely. The coverage of POLIC3-nog drops tremendously with a single exception. That POLIC3 can also excell on neural network policies can be observed for the ASNET policies. It achieves significantly higher coverage than our second baseline BRFS, indicating the ability of POLIC3 to handle even current state-of-the-art policies. This is true despite the fact that the ASNET policies are significantly larger than the FFNN ones (average number of neurons: blocks 157 (FFNN) vs. 8245 (ASNET); npuzzle 153 vs. 33990). On the other benchmarks, BRFS actually turned out competetive overall. It even achieved the highest coverage in blocks, though lagging behind in the other benchmark domains. Comparing the runtimes of POLIC3 to the two competitors PPA and BRFS (Fig. 1), the picture is even more striking. In all points below the diagonal, POLIC3

required less time to solve the instance. POLIC3 is able to improve over the competitors' runtime, often by several orders of magnitude.

## 5 RELATED WORK

**Safe reinforcement learning** Learning decision policies satisfying some safety constraints has received significant attention in safe reinforcement learning (García & Fernández, 2015; Gu et al., 2024). There are two decisive differences to our work. First, safe RL considers quantitative properties, requiring that the discounted expected value of the policy for secondary cost functions remain within some limits. Secondly, safe RL optimizes for the satisfaction of the safety constraints, but at no time guarantees that the constraints are indeed satisfied.

**Policy verification in continous environments** Policy verification has been considered especially in the context of dynamic system control (Tran et al., 2020; Tambon et al., 2022; Schilling et al., 2023; Rossi et al., 2024). In contrast to our work, they consider continuous environment models. This avoids the complications arising from discrete choices, but requires fundamentally different techniques to, e.g., deal with sets of infinitely many environment states (e.g., Fan et al., 2020).

**Policy verification in discrete environments** Policy verification for discrete action spaces has also been subject of many works. Bastani et al. (2018) present a method to obtain policies with verified performance guarantees. To this end, they synthesize a single decision tree from a deep neural network expert policy, and cast the entire policy verification problem as a single SMT. Carr et al. (2021) follow a similar idea, using standard model checkers to verify a symbolic approximation of a a given neural network policy. Both approaches provide no guarantees for the input policy. (Gross et al., 2022; 2023) verify neural network policies by implementing an interface between an off-the-shelf model checker and policy function evaluation. Policy verification however requires enumeration of the policy induced transition system, which is intractable in all but the smallest cases. Lastly, in contrast to our method, adding the support of different policy architectures in PPA (Vinzent et al., 2022; Jain et al., 2024) requires major engineering efforts.

**Other policy analysis methods** Besides verification, there has also been significant work on alternate analysis methods. Gros et al. (2023); Lampacrescia et al. (2024) consider statistical model checking methods for analyzing neural network policies. Steinmetz et al. (2022) applied techniques from software testing to spot undesired policy behavior within a symbolic environment model. Eniser et al. (2022); Mazouni et al. (2024) adopted this idea, but instead tested the policy's behavior using an environment simulator in place of a model. All these works cannot make formal guarantees about their analysis results.

## 6 CONCLUSION

We introduced POLIC3, a new policy verification algorithm, which differs from previous algorithms in its clear separation of the reasoning about the policy and the reasoning about the environment model. The former can be handled efficiently via off-the-shelf neural network and decision-tree ensemble certification tools, and the latter through standard encodings into SMT. Our experiments demonstrated that POLIC3 is more efficient and often solves more problems than state-of-the-art methods, and that it is even capable of handling policies with complex function architectures, exemplified by ASNET policies, that were previously out of reach.

That said, we have not unleashed the full power of IC3 yet. In hardware verification, a range of optimizations have been introduced that significantly boost performance, including parallelization, obligation minimization, obligation rescheduling, and reverse IC3. Our future work includes exploring those optimizations for policy verification. During our experiments, we observed that the variable order during reason minimization has a huge influence on the performance of the overall algorithm. Another highly promising direction is improving the generalization method by clever selections of the variable order. Finally, a clear limitation of policy verification so far is the need for a symbolic environment model. Through decoupling policy from environment reasoning, POLIC3 also paves the way for supporting environment models learned from data, such as those produced by model-based reinforcement learning. Exploring this potential can advance the reach of this field.

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

# A IC3 ALGORITHM

## A.1 CONNECTION TO IC3 FROM LITERATURE

Our presentation was very loosely based on that by Eén et al. (2011). Besides differences owed to the different general settings, our algorithm differs from the one in the literature in three general points. First, the original presentation of the IC3 algorithm proceeds in the **reverse direction**, attempting to build the path $s_0, a_0, \ldots, s_n$ from $\phi_S$ to $\phi_R$ backwards, i.e., starting with $s_n \models \phi_R$ and for

$i = n, \ldots, 1$, looking for the state $s_{i-1}$ that has a transition going to $s_i$. In the original setting, the directions can be exchanged easily, only requiring minor adaptations in the SAT encodings. In POLIC3, this is more difficult. In particular, the explicit search in `selectSuccessorState` is no longer possible in this manner.

In the original IC3 algorithm, the pairs $\langle s, n \rangle$ at line 12 of Alg. 1 are called **obligations**. An optimization missing in our presentation is **obligation minimization**. Instead of searching over individual states, the original IC3 algorithm considers obligations $\langle p, n \rangle$ where $p$ is a partial variable assignment, thus implicitly considering many states at once. Accordingly, `selectSuccessorState`$(p, n-1)$ returns a partial variable assignment $p'$ that (1) entails $\mathcal{F}_{n-1}$, i.e., such that $s' \models p'$ implies $s' \models \mathcal{F}_{n-1}$, and (2) all states $s$ with $s \models p$ have a transition $\langle s, a, s' \rangle \in \mathcal{T}$ so that $s' \models p'$. Importantly, to consider in the next search step such $p'$ that represent as many states as possible, `selectSuccessorState` contains an additional minimization step similar to reason generalization, which attempts to iteratively remove variable assignments from $p'$ while maintaining (1) and (2). Like above, considering partial variable assignments as obligations in POLIC3 would make the implementation of `selectSuccessorState` significantly more complicated.

Lastly, the original IC3 algorithm comes with the option to **reschedule** an obligation $\langle s, n \rangle$, inserting $\langle s, n+1 \rangle$ into the queue, when $s$ was shown to have no length-$n$ path to $\phi_R$ (line 15). Our implementation supports this, but it turned out detrimental in our experiments.

### A.2 CORRECTNESS OF GENERIC IC3 (THM. 1)

Let $\Theta = \langle \mathcal{S}, \mathcal{A}, \mathcal{T} \rangle$ be the transition system and $P = \langle \phi_S, \phi_R \rangle$ be the property to be verified. The following *frame invariants* are preserved at all time during the execution of the algorithm

**(FI)** For all $i \in \{1, \ldots, N-1\}$: $\mathcal{F}_{i+1} \subseteq \mathcal{F}_i$.

**(FII)** For all $i \in \{1, \ldots, N\}$, and all states $s \in \mathcal{S}$: if $s \models \mathcal{F}_i$, then $s \not\models \phi_R$.

**(FIII)** For all $i \in \{0, \ldots, N-1\}$, and all states $s \in \mathcal{S}$: if $s \models \mathcal{F}_i$, then $s \not\models \phi_S$.

**(FIV)** For all $i \in \{1, \ldots, N-1\}$, all states $s' \in \mathcal{S}$, and all transitions $\langle s, a, s' \rangle \in \mathcal{T}$ into $s'$: if $s' \models \mathcal{F}_i$, then $s \models \mathcal{F}_{i+1}$.

*Proof of invariants.*

- **(FI)** is satisfied as per the frame refinement step in line 17 of Alg. 1.
- **(FII)** is guaranteed by condition (i) of Assert. 4.
- **(FIII)** is guaranteed since a new frame is opened only when there is no start state $s_{\text{start}} \models \phi_S$ that satisfies $\mathcal{F}_N$ (Assert. 1), i.e., upon opening $\mathcal{F}_{N+1}$, the start states have been removed from all previous frames. Since frames are only strengthened, it is not possible that a start state gets reinserted into some frame.
- **(FIV)** is guaranteed by condition (ii) of Assert. 4.

$\square$

From the invariants, the intended property of frames follows immediately:

**Lemma 1.** *Let $n \in \{1, \ldots, N\}$, and $s_0 \in \mathcal{S}$ be such that $s_0 \not\models \mathcal{F}_n$. It holds for all paths $s_0, a_0, s_1, a_1, \ldots, s_m$ in $\Theta$ such that $s_m \models \phi_R$ that $m > n$.*

*Proof.* Proof by induction on $n$. The induction beginning, $n = 0$, holds trivially given that $\mathcal{F}_0 = \phi_R$, i.e., if $s_0 \not\models \phi_R$ then obviously there is no 0-length path from $s_0$ to $\phi_R$. For the induction step, let $s_0 \in \mathcal{S}$ be some state with a path $s_0, a_0, \ldots, s_m$ to a state $s_m \models \phi_R$ with length $m \leq n+1$. From **(FII)**, it follows that $m \geq 1$. Consider the successor state $s_1$ of $s_0$. Obviously, $s_1$ has a path with length $n$ to $\phi_R$. As per the induction hypothesis, $s_1 \models \mathcal{F}_n$. Therefore, with **(FIV)**, $s_0 \models \mathcal{F}_{n+1}$, as desired. $\square$

We are now ready to show the correctness of Alg. 1. The proof is split in three parts: correctness of the two return values, and termination.

*Proof that Alg. 1 correctly returns true.* Let $s_0, a_0, \ldots, s_N$ be the path found by Alg. 1 upon returning true. The path can be easily reconstructed by tracking for each state in the queue the transition that `selectSuccessorState` used to generate that state. In order to show that this path is indeed a witness for the satisfaction of the property, we need to show that (i) $s_0 \models \phi_S$, (ii) $s_N \models \phi_R$, and (iii) for all $i \in \{1, \ldots, N\}$, $\langle s_{i-1}, a_{i-1}, s_i \rangle \in \mathcal{T}$.

(i) is guaranteed by Assert. 1.

(ii) By Assert. 2, it holds that $s_N \models \mathcal{F}_0$ when $\langle s_N, 0 \rangle$ was inserted into the queue for the first time. Since $\mathcal{F}_0$ is not refined, $s_N \models \mathcal{F}_0$ still holds when $\langle s_N, 0 \rangle$ is popped from the queue, and at that moment the algorithm terminates.

(iii) is guaranteed by Assert. 2.

$\square$

*Proof that Alg. 1 correctly returns false.* It holds that $\mathcal{F}_N = \mathcal{F}_{N+1}$. Suppose for contradiction that there was a path $s_0, a_0, s_1, a_1, \ldots, s_m$ from some start state $s_0 \models \phi_S$ to some state $s_m$ such that $s_m \models \phi_R$. Since $s_0 \not\models \mathcal{F}_N$ (Assert. 1), it follows from Lemma 1 that $m > N$. Let $m' := m - N$. In other words, $s_{m'}$ has a path of length $N$ to $\phi_R$. Applying Lemma 1 again, it must hold that $s_{m'} \models \mathcal{F}_N$. Via **(FIV)**, it inductively follows that $s_0 \models \mathcal{F}_{N+1}$, and hence also $s_0 \models \mathcal{F}_N$. This is a contradiction to Assert. 1.

$\square$

*Proof that Alg. 1 terminates.* The inner loop (line 11) must terminate eventually, since in each step either the remaining path length counter $n$ is decremented, or a state is removed from some frame. Given that the algorithm terminates when $n = 0$ and since there are only finitely many states, both things cannot repeat forever. If the property $P$ is satisfied by $\Theta$, IC3 must eventually find a corresponding path given the correctness of the frame construction (Lemma 1) and since each start state will have to be considered eventually (Assert. 1). Assume that $P$ is not satisfied. We need to show that $\mathcal{F}_N = \mathcal{F}_{N+1}$ holds eventually. To this end, assume that $\mathcal{F}_n = \mathcal{F}_{n+1}$ holds for some $n \in \{1, \ldots, N\}$ after the call to `propagateClauses`. By Assert. 5, `propagateClauses` could have propagated every $\neg r \in \mathcal{F}_n$ into $\mathcal{F}_{n+1}$. But then, `propagateClauses` must have also propagated $\neg r$ into $\mathcal{F}_{n+2}$, and in fact into all $\mathcal{F}_m$ with $m \geq n$. In particular, $\mathcal{F}_i \subseteq \mathcal{F}_N$ and $\mathcal{F}_i \subseteq \mathcal{F}_{N+1}$. It follows from **(FI)** that $\mathcal{F}_i = \mathcal{F}_N$ and $\mathcal{F}_i = \mathcal{F}_{N+1}$, i.e., IC3 will terminate. Finally, note that such an index $i$ must exist eventually, given that there are only finitely many possible reasons. We conclude that IC3 has to terminate eventually.

$\square$

### A.3 IC3 for environment models: additional details

We provide a detailed description of the implementation of the four sub-procedures of Alg. 1 to verify whether an environment model $\mathcal{E} = \langle \textit{Vars}, \mathcal{A}, \mathcal{C} \rangle$ satisfies the property $P = \langle \phi_S, \phi_R \rangle$.

**Solving the frame transition problem** The implementation of `selectSuccessorState`, `generalizeReason`, and `propagateClauses` commonly requires a method efficiently deciding the frame transition problem (Def. 1). This can be done via SMT. Specifically, the SMT contains integer variables $\mathsf{v}_v$ for $v \in \textit{Vars}$ representing the state $s$, integer variables $\mathsf{v}'_v$ representing the state $s'$, and Boolean variables $\mathsf{c}_c$ for $c \in \mathcal{C}$ indicating the choice of the command responsible for the transition. The conditions $s \models r$ and $s' \models \mathcal{F}_n$ of (3) are mapped one-to-one into constraints of the SMT over the variables $\mathsf{v}$ and $\mathsf{v}'$ respectively. The condition that $\langle s, a, s' \rangle \in \mathcal{T}_{\mathcal{E}}$, for some $a$, is encoded as the disjunction of $\mathsf{c}_c \wedge \mathsf{guard}(c) \wedge \mathsf{effect}(c)$ over all commands $c \in \mathcal{C}$, where $\mathsf{guard}(c)$ translates the guard $grd_c$ into a constraint over the variables $\mathsf{v}$ (representing the condition $s \models grd_c$), and $\mathsf{effect}(c)$ binds the variables $\mathsf{v}'$ to the result of the application of $c$ on the values of $\mathsf{v}$ (representing the condition $s' = s[\![c]\!]$) by conjoining the constraints $\mathsf{v}'_v = \chi_v[\mathsf{v}]$ for all $v \in \textit{Vars}$, where $\chi_v = \textit{eff}_c(v)$ is the expression assigned to $v$ by the command's effect. We refer to the resulting SMT problem as $\mathsf{FrameTransition}[r, n]$.

**Start state selection** The implementation of `selectStartState` via SMT is straightforward. To find a state $s_{\text{start}} \in \mathcal{S}_{\mathcal{E}}$ such that $s_{\text{start}} \models \phi_S \wedge \mathcal{F}_N$ without enumerating the set of all states

$\mathcal{S}_{\mathcal{E}}$, we generate a constraint system with the integer variables *Vars* of the environment model, and we translate the conditions $\phi_S$ and $\mathcal{F}_N$ into constraints. The requested state can be read off of any solution of this system. If there is no solution, then $s_{\text{start}}$ does not exist. All in all, the implementation meets the specification as per Assert. 1.

**Successor state selection** With FrameTransition at hand, the implementation of `selectSuccessorState(s, n − 1)` is trivial. We solve FrameTransition$[s, n - 1]$ and reconstruct $s'$ from the solution. Assert. 2 is guaranteed by the correctness of the SMT encoding.

**Pushing clauses** Similarly, to decide in `propagateClauses` whether a clause $\neg r \in \mathcal{F}_m$ can be pushed to the next higher frame $\mathcal{F}_{m+1}$, we solve FrameTransition$[r, m]$. The clause can be pushed if the SMT is unsatisfiable. Assert. 4 again follows from the correctness of the SMT encoding. The implementation also guarantees Assert. 5 given that FrameTransition$[r, m]$ represents the condition under which $\neg r$ can be moved into $\mathcal{F}_{m+1}$ exactly, i.e., it pushes a clause forward if and only if this is possible while preserving the frame properties.

**Reason generalization** Finally, to obtain small reasons in `generalizeReason(s, n − 1)`, we follow a greedy state minimization procedure as in earlier works (Bradley, 2011; Eén et al., 2011). We initialize the reason to $r := s$. Note that this $r$ satisfies (i) and (ii) of Assert. 4 initially. (i) holds by the definition of the frames and since $n > 0$; (ii) is satisfied for each call made by Alg. 1 (line 15). We then iteratively remove individual variable assignments from $r$ while maintaining (i) and (ii). Namely, for each $v \in \textit{Vars}$, we consider $r' := r \setminus \{v \mapsto r(v)\}$. Checking whether $r'$ satisfies (i) is an easy exercise, formulated as an SMT. For condition (ii), we solve FrameTransition$[r', n]$, which has no solution exactly if (ii) is still satisfied. If we find that $r'$ satisfies both conditions, we set $r := r'$ and continue with the next variable. Otherwise, we do not change $r$ and proceed directly to the next variable. Given that (i) and (ii) remain satisfied by $r$ at all times by the design of the algorithm, this method obviously satisfies Assert. 4.

## A.4 IC3 FOR POLICY VERIFICATION: ADDITIONAL DETAILS

### A.4.1 PROOF OF THM. 2

Let $r$ be a partial variable assignment and $n \in \{0, \ldots, N\}$. Assume that (4) is satisfied, and let $s, a, s'$ be the corresponding witness, i.e., such that (i) $s \models r$, (ii) $s' \models \mathcal{F}_n$, (iii) $\langle s, a, s \rangle \in \mathcal{T}_{\mathcal{E}}$, and (iv) $\pi_\theta(s) = a$. We show that (5) is satisfied. To this end, note that the states $s$ and $s'$ satisfy (5.1) for action $a$: $s \models r$ holds by (i), $s' \models \mathcal{F}_n$ by (ii), and $\langle s, a, s' \rangle \in \mathcal{T}_{\mathcal{E}}$ by (iii). (5.2) is satisfied since $s \models r$ by (i) and $\pi_\theta(s) = a$ by (iv). This concludes the proof.

$\square$

### A.4.2 PROOF OF THM. 3

Let $r$ be a partial variable assignment, and let $\underline{A} \subseteq \mathcal{A}$ be an under-approximation of the applicable actions of the states represented by $r$, i.e., such that $\underline{A} \subseteq \mathcal{A}(s)$ holds for all $s \in \mathcal{S}_{\mathcal{E}}$ where $s \models r$. Let $s \in \mathcal{S}_{\mathcal{E}}$ be any state that satisfies $s \models r$. Assume that (6) is satisfied, i.e., that

$$f_\theta(s)[a] < \max_{a' \in \underline{A}, a' \neq a} f_\theta(s)[a'].$$

Let $\hat{a} := \pi_\theta(s)$. By the definition of $\pi_\theta$, it holds that $\hat{a} \in \mathcal{A}(s)$ and that

$$f_\theta(s)[\hat{a}] = \max_{a' \in \mathcal{A}(s)} f_\theta(s)[a'].$$

Since $\underline{A} \subseteq \mathcal{A}(s)$, in particular,

$$f_\theta(s)[\hat{a}] \geq \max_{a' \in \underline{A}} f_\theta(s)[a'] \geq \max_{a' \in \underline{A}, a' \neq a} f_\theta(s)[a'].$$

Therefore,

$$f_\theta(s)[a] < f_\theta(s)[\hat{a}].$$

Based on the definition of $\pi_\theta$, we conclude that $\pi_\theta(s) \neq a$.

$\square$

# B  EXPERIMENTS

## B.1  BENCHMARK DESCRIPTIONS

We provide a brief description of the used benchmarks. We took the models, properties, and policies from (Vinzent et al., 2022; Jain et al., 2024), and trained new policies using ASNET (Toyer et al., 2020) in two problem domains. We describe the ASNET training below. The original benchmark set contains feed-forward neural network with relu units (FFNN) and decision-tree ensemble (DTE) policies. The FFNN policies were trained using Q-learning. The FFNN policies generally had 2 hidden layers, whose size was varied in the different benchmark domains (as described below). The DTE policies were trained via imitation learning from those teacher FFNN policies considering both gradient-boosted trees as well as random forests. The size of the ensembles was generally controlled using depth limits in $\{4, 6, 8, 10, 15\}$ and number of trees in $\{5, 10, 20, 30\}$.

**beluga**  A factory logistics problem, where cargo needs to be unloaded from $n$ arriving airplanes and stored in some intermediate rack storage facilities until being requested by the production line. The start condition considers all possible orderings in which cargo can arrive. The reach condition asks whether all racks are occupied. The benchmarks vary $n \in \{4, 5, 6\}$. The FFNN policies have 2 hidden layers with $m \in \{64, 256\}$ neurons each.

**blocks**  A variant of the classic blocksworld planning problem. There are $n$ differently colored blocks which must be stacked on top of each other in a certain way. This benchmark variant comes with the additional constraint that only a limited number of blocks are allowed to be placed on the table at the same time, which is represented by the reach condition. The start state condition represents all configuration of the blocks where the constraint is satisfied. The number of blocks was ranged in $n \in \{4, 6, 8, 10\}$. The FFNN policies have 2 hidden layers with $m \in \{16, 32, 64\}$ neurons each.

**npuzzle**  Models the classic sliding tiles puzzle on a $3 \times 3$ grid. There are 8 numbered tiles and an empty tile. The tiles need to be arranged in a certain manner. The empty tile can be swapped with tiles horizontally or vertically adjacent to it. The start condition imposes a partial order over the tiles. The reach condition characterizes some unsafe tile positions to be avoided. The FFNN policies have 2 hidden layers with $m \in \{16, 32, 64\}$ neurons each.

**transport**  Models a transportation problem, where packages must be moved from left to right crossing a bridge. The truck has inertia, and can be accelerated/decelerated by one speed unit at a time. The start condition represents all states where packages are distributed arbitrary at the left side of the bridge. The reach condition asks whether the truck ever crosses the bridge with too much load. The FFNN policies have 2 hidden layers with $m \in \{16, 32, 64\}$ neurons each.

## B.2  TRAINING ASNET POLICIES

|  | blocks | npuzzle |
|---|---|---|
| Training problem sizes | $\{4, 5, \ldots, 10\}$ blocks, (25 instances in total) | $3 \times 3$ grids (30 instances in total) |
| Module layers | 2, 3 | 2, 3, 4 |
| Module dimensions | 4, 8 | 4, 8, 16, 32 |
| Activation | relu | relu |
| Weight decay | 2e-4 | 2e-4 |
| Dropout rate | 0.1 | 0.1 |
| Regularization | L1 | L1 |
| Batch size | 64 | 64 |
| Max epochs | 300 | 300 |
| Train steps | 700 | 700 |
| Policy rollout limit | 1000 steps | 1000 steps |

Table 2: ASNET training hyperparameters

We trained additional policies in blocks and npuzzle using ASNET (Toyer et al., 2020). To train a policy, ASNET requires a planning problem domain and a collection of problem instances with increasing difficulty in PDDL. To this end, we manually created PDDL encodings of blocks and npuzzle, and implemented random instance generators in python. For both domains, we let ASNET train multiple policies with different sizes. The hyperparameters are shown in Tab. 2. For each size configuration, we selected the best performing policies. The resulting generalized policies were instantiated according to the size of the models considered for verification. For npuzzle we obtained 97 additional policies in this manner. For blocks, we obtained 29 additional policies for each of the four model sizes, so 116 in total.

## B.3    ADDITIONAL RESULTS

| Benchmark | # solved | avg. $N$ | avg. runtime (s) | fraction reason generalization |
|---|---|---|---|---|
| beluga FFNN (3) | 0 | – | – | – |
| blocks FFNN (24) | 14 | 12.1 | 213.78 | 42.6% |
| npuzzle FFNN (6) | 6 | 16.3 | 4224.55 | 49.0% |
| transport FFNN (5) | 3 | 1 | 1.6 | 48.5% |
| beluga DTE (6) | 4 | 5.2 | 0.4 | 45.9% |
| blocks DTE (18) | 15 | 29.7 | 3779.72 | 76.2% |
| transport DTE (6) | 6 | 11.5 | 55.04 | 70.1% |
| blocks ASNET (116) | 96 | 7.2 | 2090.63 | 63.7% |
| npuzzle ASNET (97) | 93 | 1 | 106.61 | 53.5% |

Table 3: Per benchmark domain aggregated statistics about the POLIC3 runs. Total number of instances in braces. "# solved": number of instances solved. "avg. $N$" average path-length limit $N$ upon termination of POLIC3. "avg. runtime" runtime in seconds averaged over the solved instances. "fraction reason generalization" runtime fraction (in percent) of reason generalization from total runtime, averaged over the solved instances.

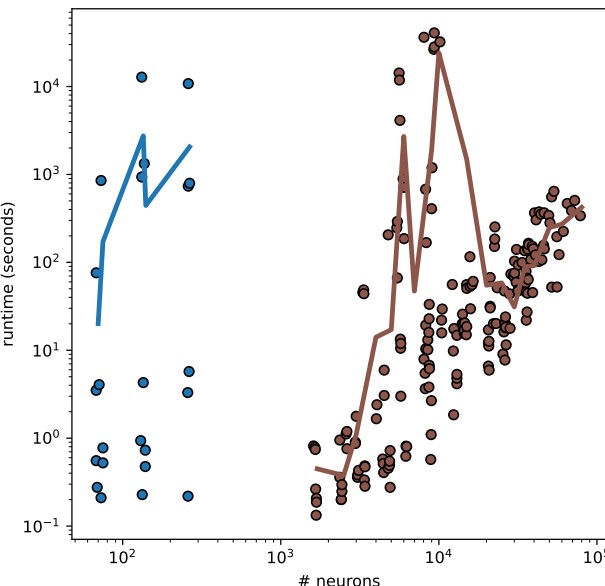

Figure 2: POLIC3 runtime (seconds) as a function of policy size for the benchmarks with neural network policies. FFNN policies in blue, ASNET in brown. Policy size is measured in number of neurons. The lines show the sliding average.

Tab. 3 and Fig. 2 show additional performance statistics for the POLIC3 runs. Runtime strongly correlates with how quickly a property satisfying path could be found. In transport (FFNN) and

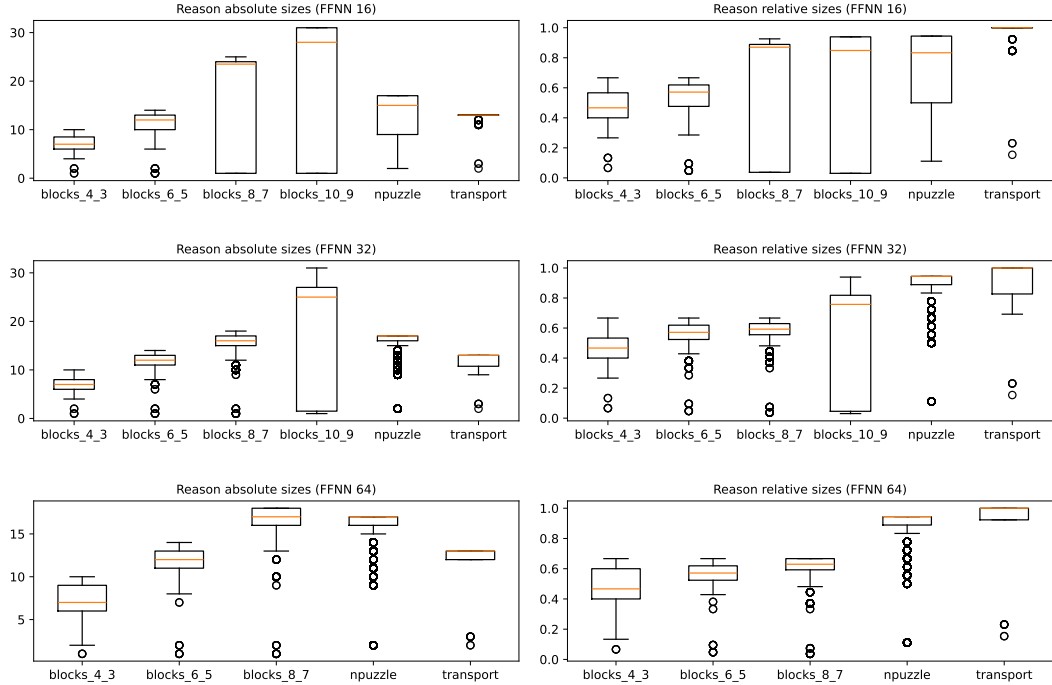

Figure 3: Per-instance distribution of POLIC3's reason sizes for the feed-forward neural network (FFNN) policies (number in braces gives the hidden layer size). Left column shows the absolute sizes. Right column shows the sizes relative to the number of variables in the JANI model.

npuzzle (ASNET) the paths were generally very short, on average a single step from some start state was enough. Runtime is comparatively small in those two domains. Comparing the FFNN and DTE policies for blocks, the data indicates that proving unsolvability of a property tends to be in general harder than showing solvability. For DTE, a satisfying path was only found in one of the 15 solved instances (in the other instances the property is unsatisfiable), compared to 5 out of 14 for the FFNN policies. In general, a big fraction of the runtime is spent on reason minimization. Similar observations were made in the original hardware verification context (cf. Eén et al., 2011). Taking a look at the runtime for neural network policies (Fig. 2), we see that the runtime generally increases steeply with the network sizes (note that the axes use log-scale). The plot also shows the importance of a *structure* of the policies. Despite being orders of magnitude larger, the ASNET policies can still be verified much more efficiently than the FFNN policies.

