# OpenReview forum: "Is my action policy safe? PolIC3 to the rescue"
_ICLR.cc/2026/Conference — Submitted to ICLR 2026_

### Official Review · Reviewer_JUAt · 2025-10-22

**Soundness:** 3
**Presentation:** 2
**Contribution:** 3
**Rating:** 4
**Confidence:** 3

**Summary:**

This work deals with policy verification.
A new method for policy verification is presented, based on the established IC3 algorithm.

**Strengths:**

* The topic is relevant.
* The idea of using and adapting a technology established in another domain is interesting.
* The abstract is written very well.

**Weaknesses:**

* The presentation seems somewhat tedious to me. A didactic revision would be desirable here. To achieve this, some details would need to be moved to the appendix.

The description is very technical. For my taste, it reads too much like a software description. I think the motivation and the individual design decisions should be explained in more detail.

Furthermore, what specific steps make the novelty should be better explained.

I am having difficulty with the presentation. I would appreciate a more detailed introduction to the problems encountered to date and the innovations presented. I would also like to see more text comparing this work with existing studies. This would be possible in a journal. In my opinion, this work, which is both highly specialized and theory-heavy, would require a complete overhaul in order to be presented as a conference paper.

Further comments:

Sometimes it says “action policy,” sometimes “policy,” sometimes “decision policies.” I think using “policy” consistently would be clearer.

“argmax” should not be italicized.

Since “a priori” is italicized, “i.e.,” “e.g.,” and “et al.” should also be italicized for consistency.

„markov“ -> „Markov“

**Questions:**

What still needs to be done so that the method can be used in a practical application?

---

> ### Author Response · Authors · 2025-11-17
>
> We thank the reviewer for the comments.
>
> We agree with the reviewer that the paper is technically dense. Nevertheless, we believe that this conference is the right venue and the right format for this work. In particular, we would hope for sparking interest on this important interdisciplinary topic and further fertilizing cross-community collaborations. Given the additional one page we will have for the camera ready, we will do our best to extend the background, and include additional illustrations and examples to make the write-up more easily digestible.
>
> Answer to the question:
>
> Scalability remains a major problem, pertaining to both the size/architecture of the policy function (e.g., neural network size) as well as to the size of the environment model. We believe that this work makes an important step forward in closing this gap. In particular, there are many possible extensions of IC3 that make us believe this is the right algorithm for this task. For example, so far, we have not made use of parallelization, hence not leveraging the full potential of modern hardware. IC3 is particularly suited to this end.

---

### Official Review · Reviewer_CaC9 · 2025-10-31

**Soundness:** 2
**Presentation:** 2
**Contribution:** 2
**Rating:** 4
**Confidence:** 3

**Summary:**

The paper studies policy safety verification for discrete environments. It adapts IC3 and proposes POLIC3, which decouples reasoning about the policy and the environment. The environment is a guarded command model; the policy is a neural net or a tree ensemble. The key step replaces an exact “policy frame transition” test with a necessary-condition surrogate that splits action feasibility and policy selection so that standard SMT handles environment reachability, and standard certification tools handle policy decisions. Experiments on several integer-variable planning benchmarks show higher coverage and faster runtimes than prior policy verification baselines, and the method can handle more complex policies such as ASNets through interface nets.

**Strengths:**

The paper writes a generic IC3 with four abstract subroutines and gives the assertions needed so that termination and correctness follow. It then maps these to environment models and to policy-constrained transition systems.

The paper explains why a direct SMT encoding of \pi is brittle, then proposes a split test with theorems that show rejection is sound: if the relaxed test fails, the exact test is false. It also gives practical constructions that reduce the policy side to LiRPA for nets and VERITAS for trees.

On known FFNN and DTE policy benchmarks, POLIC3 matches or improves solved instances and often reduces runtime, and it scales to ASNet policies through interface functions. A coverage table and per-instance runtime plots support this.

**Weaknesses:**

The core split is only a necessary condition, not equivalent to the exact policy frame test.
The method replaces the exact test with separate checks over transitions and over policy choice. Theorem 2 and Corollary 1 show only that failures of the relaxed test imply failures of the exact test, not the converse. This can produce many false positives in the relaxed test and weak pruning. The paper claims the IC3 assertions still hold, but the split gives no completeness guarantee about when refinement will cut enough states to ensure practical convergence speed. A short discussion that quantifies this gap would help.

The policy side check replaces max over applicable actions with max over an under approximation  $A\inA(s)$ that must hold for all states in r. How A is computed with SMT, how conservative it is under large guards, and what happens when guards use disjunctions or derived predicates are not discussed. Errors here directly affect the soundness of the rejection step and the strength of pruning.

For successor selection the paper enumerates commands rather than solving the approximate policy frame test, to avoid SMT complexity. This is simple but it can be expensive when branching is high. A cost bound or empirical trace would be useful to show that this choice does not dominate runtime on hard instances.

**Questions:**

For which classes of r and guards is the relaxed policy frame test equivalent to the exact test? Can you give a counterexample where it is loose and how often that arises in practice?

How do you compute A in SMT when guards include disjunctions or action precondition structure is complex? How sensitive is pruning to the size of A?

Could you share data on clause sizes and variable orders for the hard FFNN cases and show a heuristic that consistently shrinks reasons?

What guarantees do you have that F and G preserve the decisions of the original ASNet over the JANI model semantics? If none, can you bound the deviation or add a check that flags states where the mapping is ambiguous?

---

> ### Author Response · Authors · 2025-11-17
>
> We thank the reviewer for the detailed comments.
>
> We could not follow the argument why the introduced frame transition relaxation should be a weakness of the paper. This relaxation is absolutely essential to make IC3 work in our context. Moreover, as we have shown in the paper, the original frame transition conditions were relaxed carefully, ensuring that the correctness properties (i.e., soundness and completeness) of the IC3 algorithm are preserved. False positives may potentially lead to larger reasons in IC3, which does not affect the correctness properties, but could lower the performance compared to the ideal but hypothetical setting where one always obtains the smallest possible reasons. We empirically compared the relaxation to the exact test (using SMT) and observed almost identical behavior in terms of the reason computation. This observation must however be considered with caution, as the comparison was feasible only to a very limited extent due to scalability of the latter approach. We will revise the paper, making it more clear that theoretical properties of IC3 are preserved even by the relaxation. Moreover, we add a brief description of how the underapproximation $A$ is computed (cf. answer to Q2).
>
> Regarding the successor selection: Note that the SMT encoding requires representing each individual command as well, i.e., for the construction of the SMT, one also needs to iterate once over the commands. Checking the guards directly in fact has a performance advantage, as it avoid the search for a satisfying assignment done by the SMT solver. In our experiments, the reason computation was the by far most time-consuming part (up to 75% of the runtime, cf. Table 3).
>
> Questions:
> 1. The relaxed test is generally not equivalent to the exact test due to the separation of the two quantifications in (5.1) and (5.2). This holds independently of the specific form of $r$ and the guards. As a simple example where the relaxation loses precision, consider a setting with two binary variables $x1$ and $x2$
>       - States: $s1 = ( x1=1, x2=0 )$ and $s2 = ( x1=1, x2=1 )$
>       - Actions/commands:
>         - $a1$: guard $[ x1=1]$ and effect $[ x1=0 ]$
>         - $a2$: guard $[ x2=0 ]$ and effect $[ x2=1 ]$
>       - Policy $\pi(s1) = a2$ and $\pi(s2) = a1$
>       - Reason candidate $r = [ x1=1 ]$
>       - Frame $F_n = [ x1=0 \land x2=0 ]$
>
> Note that both transitions $s1 \xrightarrow{\pi(s1)} ( x1=1, x2=1 )$ and $s2 \xrightarrow{\pi(s2)} ( x1=0, x2=1 )$ violate $F_n$, i.e., the exact frame transition condition is not satisfied. The relaxation (5) is however satisfied. (5.1) is satisfied, because $s1 \models r$ and $s1 \xrightarrow{a1} ( x1=0, x2=0 )$ which satisfies $F_n$. (5.2) is satisfied because $s2 \models r$ and $\pi(s2) = a1$.
>
> 2. The computation of $A$ is relatively simple. We handle each action $a$ separately, iterating the following procedure. To check whether $a$ is applicable in all states that satisfy $r$, we simply test whether the SMT $[ r \land \bigwedge_{\text{command } c: l_c = a} \neg grd_c ]$ is satisfiable. If it is, then there exists a state that satisfies $r$ and for which none of the commands associated with $a$ are applicable, i.e., $a$ must not be in $A$. Vice versa, if it is not satisfiable, this proves that $a$ has for every state satisfying $r$ an applicable command, so $a$ is included in $A$.
>
> 3. - We have added Figure 3 to the appendix of the paper, which shows size statistics of the computed reasons for the FFNN policies. In general, one can see that the reasons are indeed larger for the more difficult instances in terms of their absolute sizes. That said, however, as far as it can be observed from the Blocksworld benchmark domain, the size/difficulty of the benchmark instances is not necessarily reflected on the relative reason size with respect to the size of the model. Unfortunately, we do not have results for other benchmark domains, since scaling (beyond policy size) benchmark instances are not available.
>     - The order used in our experiments is the order as the variables are defined in the Jani model (cf. the supp. material).
>     - Our reason computation method can be seen a heuristic minimization procedure, as we do not guarantee finding the cardinality minimal reason (which is a hard computation problem). It is not clear how the reasons could be minimized otherwise, given that there are specific requirements that the resulting reasons need to satisfy. But, we are happy to receive suggestions.
>
> 4. We made sure that $F$ is a bijection of the Jani states to the PDDL states. The ground PDDL model contains more action symbols (an artifact of PDDL; enumerating action "instantiations", like move right, move left, and so on). Our $G$ function surjectively maps the ground PDDL actions to the corresponding (high-level) Jani action. Both together ensure that the policy behavior in the PDDL model is preserved 1-to-1 when executed under the Jani model semantics.

---

> > ### Comment · Reviewer_CaC9 · 2025-11-21
> >
> > I thank the authors for their detailed response. My concerns have been addressed. Since I am not very familiar with this topic, I will consider raising my rating after discussion with the other reviewers. I have one more suggestion is that I think in the current writeup, the policy verification section is not easy to follow, especially for readers who are not familiar with the topic. Adding a concrete example would make the paper easier to read.

---

### Official Review · Reviewer_ujuV · 2025-11-03

**Soundness:** 3
**Presentation:** 3
**Contribution:** 2
**Rating:** 4
**Confidence:** 3

**Summary:**

The paper introduces a policy verification method based on the IC3 algorithm. The algorithm answers the reachability question - that is, whether a particular state is reachable in at least one of the executions of the policy.
Essentially, the paper suggests to treat policy verification as a model checking problem for a safety (reachability) property.

**Strengths:**

The approach appears novel - at least I didn't see any previous work with an overlapping contribution. The paper is well-written, and the claims about efficiency are believable. The approach is implemented and demonstrates good experimental results.

**Weaknesses:**

The theoretical contribution is weak: it consists of applying an existing algorithm to a known problem.

Moreover, the claim of solving policy verification with IC3 overstates the contribution somewhat, as the paper presents verification of reachability properties of a policy only.

**Questions:**

1. Can you use a SAT solver for this problem as well (performance problems notwithstanding)?
2. Can you verify liveness properties?
3. Traditionally, verification of policies is probabilistic. It seems that your approach cannot provide probabilistic guarantees, is that correct?

---

> ### Author Response · Authors · 2025-11-17
>
> We thank the reviewer for the feedback.
>
> Before answering the questions, we'd like to point out that while the IC3 algorithm is well known, making it work for policy safety verification is absolutely non-trivial and there lies exactly our contribution. Our work includes in particular approximations of different IC3 subcomponents, alongside with theoretical proofs that those approximations preserve correctness (soundness and completeness) of the algorithm. Applying the plain IC3 method to our problems does not scale at all. In our experiments, this variant could not even handle the smallest instances.
>
> We will revise the text to make it more clear that we deal with reachability properties specifically.
>
> Answers to the questions:
>
> 1. One can encode the problem of whether there is a path from $\phi_S$ to $\phi_R$ with length $n$ as a single SMT. By iterating over $n$, one can therewith check the satisfaction of the property. This is known as Bounded Model Checking. This method however has significant downsides, especially in our setting. For each of step limit $n$, it requires encoding the transition relation as well as the whole policy action selection (i.e., the complete neural net / tree ensemble) $n$ times. Even for $n = 1$, this is often already infeasible. We refer to the experiments in [1], which evaluated this method.
>
> 2. There are techniques to handle liveness properties in IC3 [2], which can in principle be applied to our variant directly. Our implementation does currently however only support reachability properties.
>
> 3. This is correct, IC3 per se is a qualitative verification algorithm. There however exist extensions of this method to deal with quantitative (probabilistic) properties [3], and those extensions can presumably be applied to our context as well.
>
> References
>
> [1] Vinzent, M., Sharma, S., & Hoffmann, J. (2023). Neural policy safety verification via predicate abstraction: CEGAR. In Proceedings of the AAAI Conference on Artificial Intelligence (AAAI).
>
> [2] Claessen, K., & Sörensson, N. (2012). A liveness checking algorithm that counts. In 2012 Formal Methods in Computer-Aided Design (FMCAD). IEEE.
>
> [3] Batz, K., Junges, S., Kaminski, B. L., Katoen, J. P., Matheja, C., & Schröer, P. (2020). PrIC3: property directed reachability for MDPs. In International Conference on Computer Aided Verification (CAV).

---

### Meta-Review · Area_Chair_bqsQ · 2026-01-04

**Summary:**

The reviewers mentioned that the contribution is limited. Furthermore, multiple concerns regarding the presentation were raised. The paper could undergo significant revision to clarify its contribution and make it more accessible.

Need substantial improvements before resubmission.

**Reviewer Concerns:**

- `ujuV` found the contribution limited at the same time that the claims of solving the policy verification were too strong. The rebuttal argues that the contribution is significant due to the development of subcomponents (which are not specified) that are necessary to policy verification. Furthermore, it states that the text will be revised to soften the paper's claims. In the end, the rebuttal was vague and likely would not change the reviewer's assessment.
- `CaC9` had questions regarding the core split and successor selection mechanisms; the rebuttal was sufficient to clarify these questions.
- `JUAt` mentioned that the presentation could be improved. The rebuttal promised to improve the presentation in the camera-ready version but did not make any changes to the submission, so it is unclear how the paper would change in the final version.

**Reviewer Scores:**

- `ujuV`: 4 -> 4
- `CaC9`: 4 -> 6
- `JUAt`: 4 -> 4

---

### Decision · Program_Chairs · 2026-01-26

Reject